

# Eduvy
## Integrated Remote Tutoring Platform

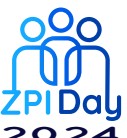

**Autors**: Wojciech Koryś ⓘ · Konrad Rudnicki ⓘ · Wojciech Skuła ⓘ · Filip Szydłak ⓘ

**Supervisor:** Marcin Jodłowiec

### Abstract

Eduvy is a unified remote tutoring platform designed to streamline the connection between students and tutors. By offering features such as lesson booking, a built-in calendar, online meeting integration, secure payments, tutor browsing, filtering, and chat, Eduvy eliminates the need for multiple tools, providing a seamless and efficient solution for online education. The objective of the platform is to simplify the tutoring process, making it more accessible and user-friendly while ensuring data privacy and security. Eduvy addresses the common challenges of finding and booking tutors by centralizing all aspects of the tutoring experience on a single platform. With a modern responsive design, Eduvy improves accessibility, efficiency, and overall learning experience. Its innovative approach aims to transform online education, offering significant value to students and tutors alike and redefining how lessons are planned and conducted in the digital age.

## 1   INTRODUCTION

The online tutoring landscape is riddled with inefficiencies, primarily due to the reliance on fragmented tools for lesson scheduling, communication, and payment management. This makes it difficult for the student to find the right tutor while limiting the tutor in their workflow. Such thoughts led to the development of Eduvy as a remote tutoring platform where finding a suitable match and managing it into a single, user-friendly product would be possible.

Eduvy aspires to create a seamless platform for students where finding, booking, and conducting lessons with tutors, and for tutors, managing their availability, communicating with students, and receiving secure payments can be availed in one application. From integrating a built-in calendar and online meeting tools to providing secure payment gateways and an intuitive tutor search with advanced filtering options, Eduvy seeks to revolutionize online tutoring. By focusing on data privacy, security, and responsive design, it is sure to develop a robust and reliable solution for an ever-growing online education market.

Eduvy's main goal is the enrichment of the tutoring process both for students and tutors; in this regard, it eliminates inefficiency and consolidates all the important tools into one scalable application. Built on a microservices architecture using Spring Boot, Eduvy guarantees scalability for a wide user base; meanwhile, the integration of Auth0 with OAuth2 standards ensures secure and private data handling. The front-end, built with Vue.js, provides a very responsive and user-friendly interface, ensuring smooth navigation across devices. Some features are developed to make the process of finding the right tutor very easy and smooth: browsing and filtering tutors, for example.

The general objectives of the Eduvy project were the development of easily accessible tutoring services, the fastening of the student-to-tutor matching process, and the creation of a professional and engaging online platform that meets the needs of modern users. By providing an all-in-one solution, Eduvy enhances the tutoring session experience by creating better communication, time management, and productivity. This makes things easier for students while making learning effective and yet easy, while supporting tutors by keeping things less complicated in workflow and reach for them. For the future, it has huge potential to facilitate online tutoring easily, effectively, and for everyone, which will not be limited to a single stratum.

In this article, we first review the current state of online tutoring platforms and identify the gaps that Eduvy aims to address. This is followed by a comparison of Eduvy with platforms like E-korepetycje and Superprof, highlighting its unique, integrated approach.

Next, we explore the architecture and technology stack of Eduvy, including its microservices design and key features such as lesson scheduling, tutor filtering, secure payments, and communication tools. This section also covers the challenges faced during development and the solutions implemented to deliver a scalable and user-friendly platform.

We then present the results, showcasing the functionality of Eduvy through examples and demonstrating how it meets its objectives. Finally, the conclusion summarizes the impact of Eduvy on online tutoring and outlines future directions, including AI integration ensuring continued innovation.

## 2   RELATED WORK

Eduvy aims to offer simplicity in the online tutoring experience, filling various gaps in existing platforms, such as E-korepetycje or Superprof. Although E-korepetycje provides a marketplace for tutors and students, it does not allow scheduling and payment through the website. It also does not support online lessons or appointment management on the platform. Similarly, Superprof has an extensive tutor directory, but uses other tools for communication and lesson management, creating friction for users. Eduvy stands out by offering a completely integrated solution that combines lesson booking, scheduling, chat, and secure payments into one easy-to-use platform.

The following table (Table 1) highlights the key functionalities of Eduvy compared to E-korepetycje and Superprof, showcasing how Eduvy addresses these gaps with its integrated solution:

Table 1: Comparison of Functionalities Between Eduvy, E-korepetycje, and Superprof

| Functionality | Eduvy | E-korepetycje | Superprof |
|---|---|---|---|
| Online Tutor Marketplace | Yes | Yes | Yes |
| Lesson Booking | Yes | No | Yes |
| Scheduling | Yes | No | No |
| Secure Payments | Yes | No | Yes |
| Integrated Chat | Yes | No | Limited (for accepting the lesson) |
| Integrated Meetings | Yes | No | No |
| Appointment Management | Yes | No | Limited |
| User Authentication and Privacy | Yes | Limited | Limited |
| Responsive and Interactive UI | Yes | No | Yes |

Eduvy's technology stack has been carefully selected to balance scalability, security, and user experience with resources and time. The architecture is microservice-based to allow for modular, independently deployable services that are easier to maintain, scale, and add features to in the future. Java with Spring Boot forms the backbone of the back-end, a mature ecosystem that efficiently handles complex business logics. The configured setup reduces overhead during development, while robust security frameworks ensure data privacy is guaranteed.

The front-end is implemented with Vue.js, using TypeScript and Tailwind CSS for an interactive and responsive interface. Vue.js brings simplicity and reactivity, TypeScript offers code maintainability and reduces errors, and Tailwind CSS allows designers to develop fast and consistent designs. PostgreSQL was selected as the database for its advanced querying capabilities, strong data integrity, and compatibility with microservices, providing a reliable and scalable storage solution.

These technologies align with the team's expertise, address resource and time constraints while laying the foundation for a user-focused, secure platform that can grow with the project.

With the time constraint, the team had to focus on core features such as booking and scheduling lessons, leaving iteration for advanced functionalities like online meeting and payment integrations. Resource limitations influenced the decision to focus on open source technologies and frameworks that align with the team's expertise. The challenges included optimizing communication between microservices, maintaining a secure user authentication system, and ensuring smooth integration of front-end and back-end components. Eduvy does a fair job in overcoming these limitations of becoming the complete, easy-to-use, safe online tutoring destination with great visibility among other solutions.

## 3   RESULTS

Eduvy's initiative was carried out to successfully develop a comprehensive platform that enhances the tutoring experience for both learners and educators.

As a result, following snippets from the application are presented:

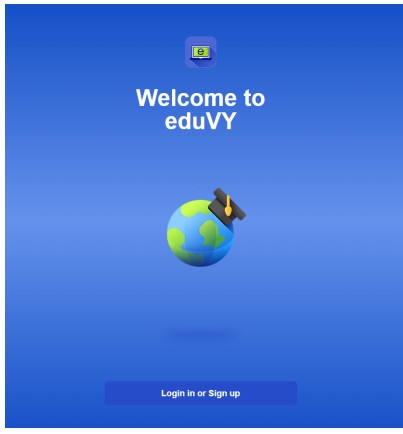

Figure 1: Login and Register Page

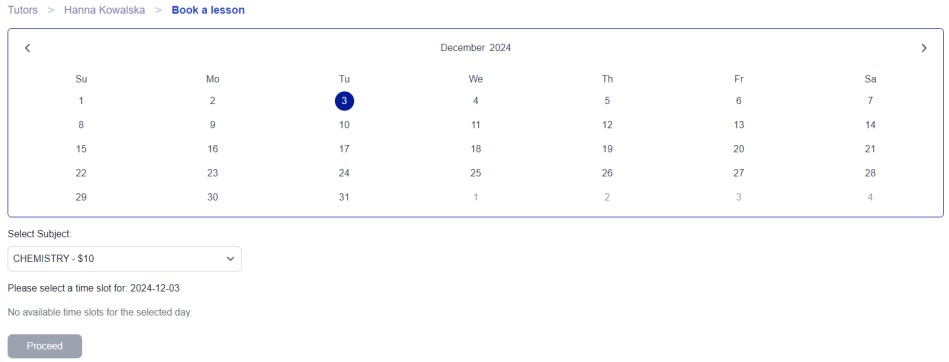

Figure 2: Calendar Component

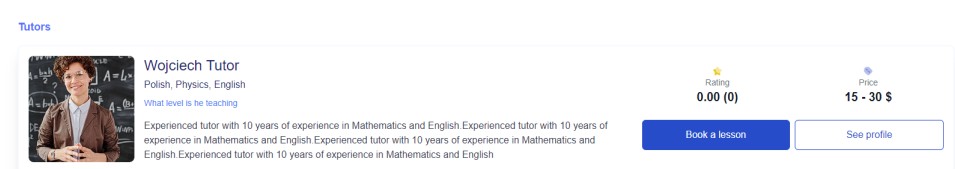

Figure 3: Tutor Profile

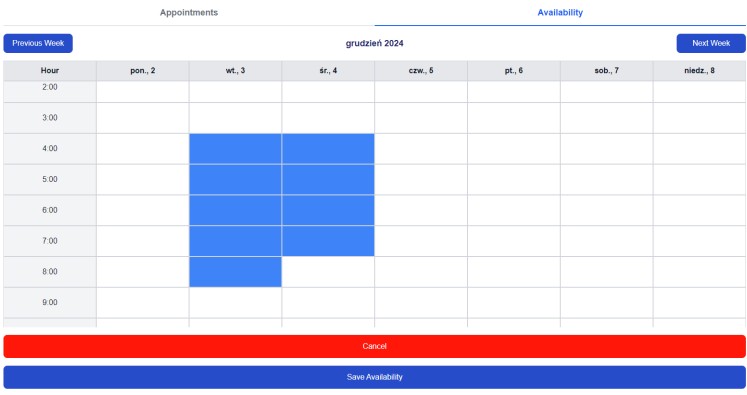

Figure 4: Availability Component

The platform includes critical features such as secure authentication through Auth0, the ability to search for and filter tutors, an attractive calendar interface, lesson scheduling, payment processing, integration with Google Meet, and a real-time communication system. Built on top of a scalable microservices architecture, this platform employs Spring Boot for the back-end and Vue.js with Tailwind

CSS for the front-end. Each microservice is designed with a dedicated PostgreSQL database to ensure data isolation and scalability, enabling efficient and independent management of data. This architecture allows each service to be deployed, scaled, and maintained individually, offering flexibility, high performance, and resilience for the application.

Eduvy provides a seamless user experience across devices. The platform makes all aspects of tutoring, from finding and booking lessons to managing payments and communication, much easier by consolidating them into one efficient application. Eduvy helps students efficiently by providing a greater choice of tutors, better time management capabilities, and convenience. For tutors, the platform relieves administrative burdens, expands their reach, and smooths operational flows. By following OAuth2 standards for secure data management and using Docker to deploy the app on an Ubuntu server, Eduvy meets its objective of setting up a tutoring platform that is accessible, professional, and engaging.

This will be an ongoing objective: to transform online tutoring through better access, communication, and productivity, all while making it possible to scale to a diverse international audience.

# 4    CONCLUSION

Eduvy embodies a transformative approach to online tutoring, addressing inefficiencies and challenges by integrating key functionalities into a single, seamless platform. By consolidating lesson booking, scheduling, communication, and payment processing, Eduvy eliminates the fragmented experience caused by multiple disconnected tools. Its foundation on a robust microservices architecture ensures scalability and reliability, meeting both current demands and future needs.

For students, Eduvy simplifies the process of finding and working with qualified tutors, enhancing learning outcomes through better time management, organized communication, and secure transactions. For teachers, it reduces administrative overhead, broadens their reach to new audiences, and provides an intuitive system to manage their schedules and interactions. These benefits create a balanced and efficient ecosystem for both parties, ensuring a smooth and enjoyable tutoring experience.

## 4.1    Conclusions

Eduvy helps address inefficiencies in online tutoring by offering a one-stop solution to consolidate lesson booking, scheduling, communication, and payment processing through a secure and seamless application. Powered by a highly scalable microservices architecture, Eduvy offers reliability for present and future growth, while its user-oriented design means intuitive experiences across devices. Some of the core feature integrations include Autho for secure authentication, integration to Google Meet, chat functionality, and a calendar interface, which really enhances its functionality and usability.

Undoubtedly, the major achievement of this project has transformed online tutoring into an easy, efficient, and organized affair. The platform simplifies the search for eligible tutors and improves learning due to effective time management. It means less administrative hassle for tutors and, therefore, the possibility of reaching a wider audience. Indeed, Eduvy's technical achievements-secure data handling and a responsive interface-mean a robust, professional experience that makes it one of the leading solutions in the education technology market.

What truly sets the platform apart is where a connection is made between students and tutors, building better communication, productivity, and paving the way to newer developments such as AI personalization and mobile application development. With Eduvy, the level bar of quality in the online tutoring niche reaches new heights, with multiple promises for audiences of interest of both technical and business issues.

## 4.2    Future Directions

As Eduvy evolves, integrating artificial intelligence (AI) offers a transformative opportunity to enhance its impact and scalability. AI can improve the student-tutor matching process by analyzing learning preferences, tutor expertise, and performance data, providing personalized recommendations for better learning outcomes. In addition, AI-driven analytics can provide tutors with valuable insights to refine their teaching methods, enriching the overall learning experience.

Future AI advancements could include predictive tools to identify students at risk of falling behind and suggest tailored interventions. Intelligent scheduling systems could also streamline lesson planning by optimizing timings based on user behavior and preferences. By focusing on AI integration, Eduvy can position itself as an innovative leader in the online education market, ensuring a smarter and more personalized tutoring experience for its users.

## 4.3 Acknowledgments

We express our sincere gratitude to our supervisor, dr inż. Marcin Jodłowiec, for his invaluable guidance and support throughout the development of this project. His expertise and feedback were crucial in shaping Eduvy into a comprehensive and innovative platform.

We also extend our heartfelt thanks to dr inż. Marcin Maleszka and dr inż. Rafał Palak for their insightful advice and assistance, which significantly contributed to the progress and refinement of our work.

Additionally, we are grateful to dr inż. Krystian Wojtkiewicz and dr inż. Wojciech Thomas for their dedication and efforts in organizing ZPI Day, providing us with the opportunity to present our project and gain valuable feedback from peers and experts alike.

Thank you for your unwavering support and encouragement.

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
