# OpenReview forum: "Eduvy"
_pwr.edu.pl/Wrocław_University_of_Science_and_Technology/2024/ZPI_Day — Wrocław University of Science and Technology 2024 ZPI Day Submission_

### Official Review · Reviewer_VdAB · 2024-12-03
**The review**

**Confidence:** 5
**Significance Of Results:** 4
**Overall Quality:** 4

**Compliance With Template:**

5: Very High Quality – The article contains all the required sections, which are written in a very detailed, clear, and error-free manner. The structure is professional and meets expectations, and the content adheres to the highest substantive and formal standards.

**Description Of Results:**

4: High Quality – The results are described in detail and supported by usage examples or evaluations. The description is reliable but may lack full depth of analysis.

**Feedback On Consistency:**

The paper is consistent and meets all the requirements for a technical report. It is clear and easy to read. However, one notable shortcoming is the lack of details regarding the development process. While the technologies and architecture are thoroughly described, the report does not provide information about the workflow methodology (e.g., Agile) or the project management tools employed. Additionally, including key performance metrics would enhance the evaluation of the project's success. Metrics such as response times, system uptime, or user satisfaction could be valuable for measuring the quality and effectiveness of the solution. Moreover, Figures 1, 2, and 3 are not referenced in the text, which limits their contextual relevance and impact.

**Potential For Development:**

The future work description is focused on application of AI solution, which is conformant to the current market trends.

**Project Nature Evaluation:**

The project presents clear characteristics of an engineering work through its focus on solving real-world business problems in online tutoring. The application of advanced technical means, such as service-like architecture, OAuth2-based security, and responsive design using popular frameworks like Spring Boot and Vue.js, ensures scalability and usability.
However, the project has potentially some shortcommings: the paper lacks evidence of thorough testing, such as unit tests, end-to-end (E2E) tests, or usability testing, which are critical to validating the platform's reliability.

**Technical Language Precision:**

4: High Quality – The language is appropriate for a technical report. Terminology is used correctly, and statements are precise, with only minor shortcomings that do not affect the overall clarity.

---

### Official Review · Reviewer_5gLv · 2024-12-04
**Eduvy**

**Confidence:** 5
**Significance Of Results:** 3
**Overall Quality:** 4

**Compliance With Template:**

4: High Quality – The article contains all the required sections, which are well-written and substantively correct, although minor errors or shortcomings may be present. The overall structure is clear and coherent.

**Description Of Results:**

3: Average Quality – The results are described with moderate detail. Some examples or evaluation elements are present but insufficiently developed or incomplete.

**Feedback On Consistency:**

Wątpliwości budzi zakres realizacji. Planowany jest szerszy niż zrealizowany (co jest nieco w tekście ukryte). Planowany obejmuje m.in. spotkania on-line i zintegrowane płatności. W zaimplementowanym produkcie tych funkcji nie ma, co sprawia, że jego innowacyjność, w stosunku do konkurencji spada. Interfejs aplikacji jest brzydki. Literatura bez kompletnych źródeł.
Mam wrażenie, że część artykułu została wygenerowana.

**Potential For Development:**

Tak; wskazano możliwości zastosowania AI do łączenia uczniów i nauczycieli.

**Project Nature Evaluation:**

Aplikacja 3 warstwowa; zaimplementowano podstawowe funkcjonalności.

**Technical Language Precision:**

5: Very High Quality – The language is entirely appropriate for a technical report. All terms are used correctly and precisely, and the style is professional, clear, and coherent, without any errors or ambiguities.

---

### Official Review · Reviewer_uRf9 · 2024-12-06
**GenAI z pierwszego prompta**

**Confidence:** 4
**Significance Of Results:** 2
**Overall Quality:** 3

**Compliance With Template:**

3: Average Quality – The article includes most of the required sections, but some may be incomplete, written in a general or unclear manner. The content is correct but requires further refinement.

**Description Of Results:**

1: Very Low Quality – The results are either not described or described in a minimal, unclear manner, without any examples or evidence. No evaluation is provided.

**Feedback On Consistency:**

Typowe, redundantne lanie wody w stylu GenAI, z którego nic nie wynika, a na pewno nie potwierdza istnienia opisanego produktu - co najwyżej pomysł. Np. słowo "seamless" występuje tu 5 razy, zawsze w tym samym, nietechnicznym kontekście.

**Potential For Development:**

Ogromny potencjał, bo opisany jest jedynie pomysł, nie realizacja.

**Project Nature Evaluation:**

Całość brzmi jak tekst z broszury reklamowej, tylko bardziej redundantnie (w kółko te same marketingowe frazesy).
References to nie bibliografia (nie mówiąc o braku odniesień do niej z tekstu), a co najwyżej shopping list

**Technical Language Precision:**

3: Average Quality – The language is mostly appropriate but may contain minor terminological or stylistic errors. Some statements might lack precision or require improvement for better readability.

---

### Decision · Program_Chairs · 2024-12-10

Accept (Poster)